# Transgenesis and web resources in quail

Olivier Serralbo[1]*, David Salgado[2], Nadège Véron[1], Caitlin Cooper[3], Marie-Julie Dejardin[4], Timothy Doran[3], Jérome Gros[5]*, Christophe Marcelle[1,4]*

[1]Australian Regenerative Medicine Institute (ARMI), Monash University, Clayton, Australia; [2]Marseille Medical Genetics (GMGF), Aix Marseille University, Marseille, France; [3]CSIRO Health & Biosecurity, Australian Animal Health Laboratory, Geelong, Australia; [4]Institut NeuroMyoGène (INMG), University Claude Bernard Lyon 1, Lyon, France; [5]Department of Developmental and Stem Cell Biology, Pasteur Institute, Paris, France

**Abstract** Due to its amenability to manipulations, to live observation and its striking similarities to mammals, the chicken embryo has been one of the major animal models in biomedical research. Although it is technically possible to genome-edit the chicken, its long generation time (6 months to sexual maturity) makes it an impractical lab model and has prevented it widespread use in research. The Japanese quail (*Coturnix coturnix japonica*) is an attractive alternative, very similar to the chicken, but with the decisive asset of a much shorter generation time (1.5 months). In recent years, transgenic quail lines have been described. Most of them were generated using replication-deficient lentiviruses, a technique that presents diverse limitations. Here, we introduce a novel technology to perform transgenesis in quail, based on the in vivo transfection of plasmids in circulating Primordial Germ Cells (PGCs). This technique is simple, efficient and allows using the infinite variety of genome engineering approaches developed in other models. Furthermore, we present a website centralizing quail genomic and technological information to facilitate the design of genome-editing strategies, showcase the past and future transgenic quail lines and foster collaborative work within the avian community.

**\*For correspondence:**
olivier.serralbo@monash.edu (OS);
jgros@pasteur.fr (JG);
christophe.marcelle@monash.edu (CM)

**Competing interests:** The authors declare that no competing interests exist.

## Introduction

Due to the easy access of chicken embryos to manipulation, this model has been at the origin of numerous seminal discoveries in a diverse range of topics (e.g. immunology, genetics, virology, cancer, cell biology, ethology, etc.; *Stern, 2005*). The specificities of the avian model have fostered the development of efficient techniques to target cells and tissues (e.g. in vivo electroporation, lipophilic dye labeling) that, combined with high-end imaging technologies (e.g. light sheet and fast-scanning two-photon excitation microscopy), have allowed the studies of dynamic morphogenetic processes in an amniote embryo environment with exceptional spatiotemporal resolution. Genetic approaches in birds have, however, lagged behind the two main genetic vertebrate models (mouse and fish), largely due to the particularities of the reproductive physiology of birds. The zygote is very difficult to access as it initiates its development internally, in the hen's oviduct and on a large yolk. By the time the egg is laid, the embryo has already developed into a blastoderm of about 40,000–50,000 cells with the germ line lineage already set aside (*Eyal-Giladi and Kochav, 1976*; *Intarapat and Stern, 2013*). Because of this, most researchers in avian genetics have focused their efforts on two distinct methods (*Nishijima and Iijima, 2013*): i) the genetic manipulation of primordial germ cells (PGCs) in vitro, which are injected back into recipient embryos (*Idoko-Akoh et al., 2018*; *Park et al., 2014*; *Taylor et al., 2017*; *van de Lavoir et al., 2006*) or ii) the direct infection of PGCs within the subgerminal cavity with replication-defective lentiviruses (*Bosselman et al., 1989*; *McGrew et al., 2004*). Both approaches have been applied successfully to chicken (*Nishijima and Iijima, 2013*). However, due to their long generation time (6 months to sexual maturity), transgenesis in chicken is

impracticable within the timeframe of most research projects. Therefore, this technology has not been widely used in basic research and its development has been mainly driven by industrial interests to produce biologically active pharmaceutical proteins in eggs (*Lillico et al., 2007*; *Nishijima and Iijima, 2013*; *Woodfint et al., 2018*).

Japanese quail (*Corturnix coturnix japonica*) is a better alternative for avian genetics. Quail is a well-known model in the developmental biology field through its extensive use in the so-called quail-chick chimera technique (*Le Douarin, 1973*; *Le Douarin, 2005*). Indeed, quail and chicken are close relatives (Order: Galliformes; Family: Phasianidae), they share on average 95% homology at the gene sequence level (GenBank) and quails are susceptible to most chicken diseases (*Barnes, 1987*). Similar to chicken, quail lay about one egg per day, and their smaller size makes them more compatible with the frequently limited space of animal facilities. A decisive advantage of quail over chicken is that they reach sexual maturity in 6 weeks. Thus, their life cycle is significantly shorter than that of chicken (26 weeks), but also of mice (8 weeks) or zebrafish (12 weeks). As no reliable technique for the culture of quail PGCs has yet been described, transgenesis in this species has mainly relied on the infection of blastoderm stage embryos (or circulating *Zhang et al., 2012*) with lentiviruses carrying fluorescent markers under ubiquitous or tissue-specific promoters, leading in recent years to the generation of a few quail lines: ubiquitous (*Huss et al., 2015*; *Zhang et al., 2012*); neuronal (*Scott and Lois, 2005*; *Seidl et al., 2013*; adipose *Ahn et al., 2015*; intestinal *Woodfint et al., 2017*; endothelial *Sato et al., 2010*). A high titre of lentivirus is required to succeed at producing chimeric embryos carrying transgenic PGCs using this technique (*Poynter and Lansford, 2008*; *Scott and Lois, 2005*). Even though large inserts (more than 10 kilobases) can theoretically be packaged using lentiviruses, a sharp drop in viral titre is observed with insert above 4–5 kb, likely representing the upper limit of constructs that can be used to generate transgenic birds (*Kumar et al., 2001*; our observation). An important advance in the field was the recent use of adenoviral vectors to perform the first CRISPR/Cas9-mediated targeted gene knockout in quail by direct infection of blastoderm cells (*Lee et al., 2019*). Whereas viruses should be sufficient for a number of applications, the size limitation of inserted constructs may be a significant restriction for their use to generate complex transgenes.

For biosafety and commercial reasons, non-viral and simpler technologies for achieving transgenic poultry have been tested. The direct in vivo transfection of PGCs with commercially available liposome-based transfection reagents has recently been successfully used to generate transgenic chicken lines (*Cooper et al., 2018b*; *Cooper et al., 2018a*; *Tyack et al., 2013*). Insertion of Tol2 (T2) transposable elements in constructs allows the efficient and stable integration of foreign DNA into the genome of transfected PGCs in the presence of exogenously provided transposase protein. Since DNA inserts up to a size of 11 kb can be cloned between Tol2 sequences with no visible loss of transposable efficiency (*Kawakami, 2007*) this allows using larger and/or more complex constructs than in viruses. Another advantage of transgenesis using the Tol2 transposon system is that it leads to minimal epigenetic silencing during development (*Macdonald et al., 2012*).

Here, we used the direct transfection of PGCs in the bloodstream of quail embryos. We present three novel quail transgenic lines carrying fluorescent proteins under ubiquitous and tissue-specific promoters. Illustrating the flexibility of the plasmid-based system, we integrated a cassette containing the GFP protein driven by a promoter for Chick βB1-Crystallin (*Taube et al., 2002*) making the identification of transgenic animals possible at hatching by UV illumination. Moreover, we present a community website where existing lines are described and new ones can be uploaded. This portal also centralizes useful technical resources for the study of this model and genomic information, based on the recently sequenced quail genome that will facilitate the generation of novel lines and the design of functional experiments in quail.

## Results

### Direct transfection of quail PGCs

Unlike mammals, avian PGCs temporarily transit through the blood system. Chicken PGCs initially located at the end of gastrulation within the germinal crescent, enter the extra-embryonic blood vessels at about 30 hr of incubation (E1.5, HH9) and begin to circulate throughout the embryo. Their number in the blood peaks around E2/2.5 (HH15-16). By E3 (HH20), PGCs actively migrate back in

the embryo and into the gonad anlagen (*Nakamura et al., 2007*; *Nieuwkoop and Sutasurya, 1979*). Chicken PGCs are transfected during their transient journey in the bloodstream, using a transfection mix containing lipofectamine, a transgenesis plasmid with Tol2 elements flanking the DNA construct to be inserted and a plasmid coding for transposase under an ubiquitous promoter (*Tyack et al., 2013*).

To test whether the direct transfection of PGCs can be achieved in quails, we determined whether we could observe PGCs transfected with fluorescent proteins once they colonized the gonads. A transfection mix containing a transgenesis plasmid coding for a membrane-localised GFP (GFP-CAAX) and a nuclear monomeric Cherry (NLS-mCherry) under the control of a strong ubiquitous promoter (CAG: CMV promoter, chick beta actin enhancer; *Niwa et al., 1991*) was injected

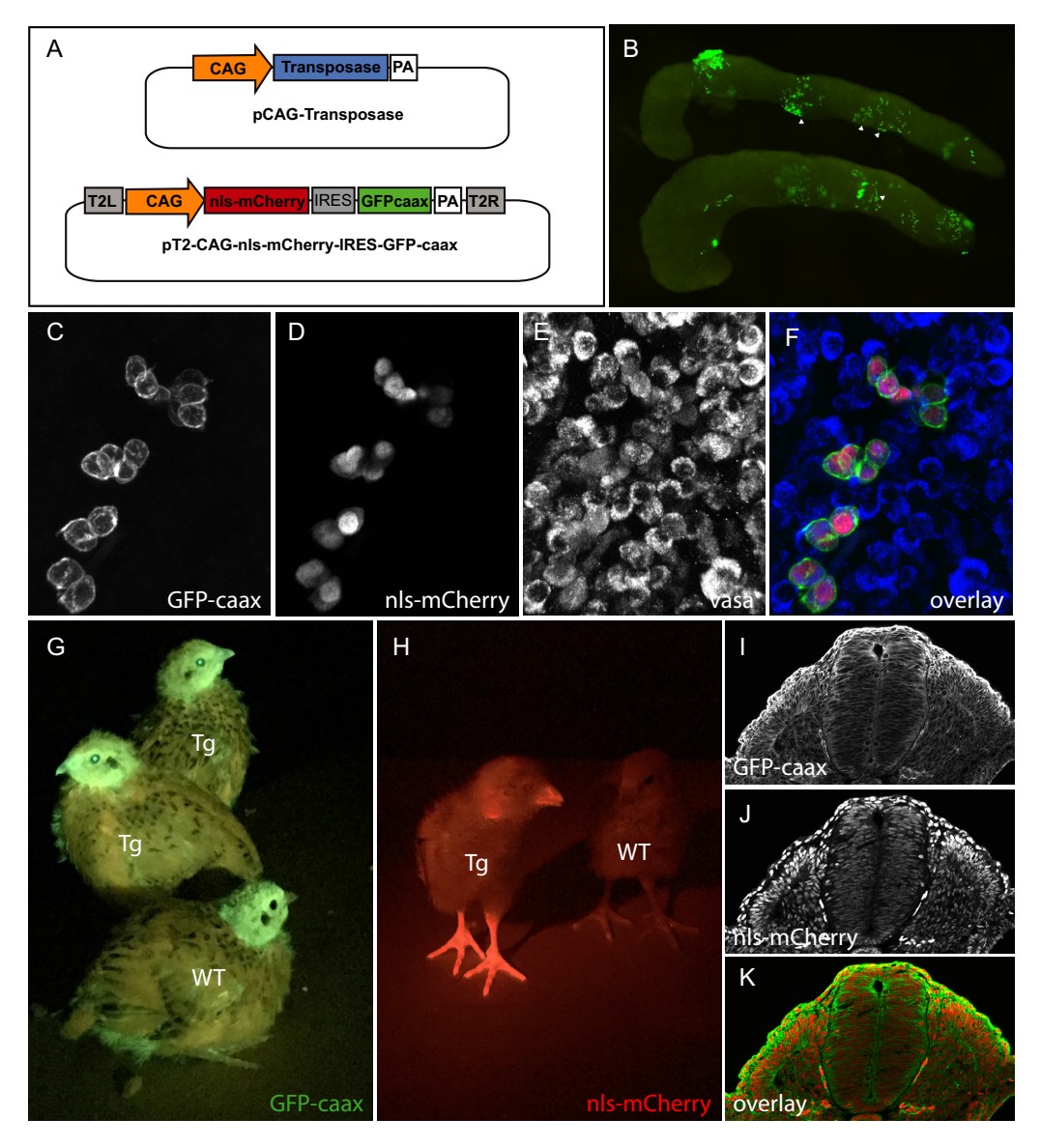

**Figure 1.** PGC transfection in vivo and generation of a quail line ubiquitously expressing membranal GFP and nuclear mCherry. (**A**) Vectors used in the injection mix. (**B**) Gonads from E7 embryo dissected 5 days after in vivo PGC transfection, showing GFP-positive transfected PGCs (arrowheads). (**C–F**) Confocal views of transfected (GFP- and mCherry-positive) PGCs among non-transfected PGCs, in the gonads of E7 injected embryo. PGCs are recognized by their expression of the Vasa marker (**E,F**). (**G–H**) Transgenic (Tg) and wild-type (WT) chicks showing ubiquitous expression of membranal GFP and nuclear mCherry when observed with UV goggles. (**I–J**) cross-section of an E3 transgenic quail embryo, showing strong and ubiquitous expression of the transgenes in all cells of the embryo.

into the bloodstream of quail embryos (*Figure 1A*). Quail develop slightly faster than chicken (17–18 days for quails; 21 days for chicken). However, we found that the timing of injection of the transfection mix resulting in the colonization of fluorescent PGCs was similar to that determined in chicken, that is at 2 days (E2) of quail embryonic development. Indeed, 5 days after injection (at E7), we observed that gonads contained many fluorescent cells (*Figure 1B*). We confirmed that these were PGCs using whole-mount immunostaining with a PGC-specific marker VASA (*Figure 1C–F*).

## Generation of a lens-specific GFP minigene to facilitate the selection of transgenic birds

To facilitate the selection of transgenic (F1) birds, we devised a fluorescent selection marker readily visible in the lens at hatching under blue light illumination. We isolated a 462bp-long lens-specific promoter of the βB1crystallin gene (CRYBB1; *Duncan et al., 1995*) from chicken genomic DNA and cloned it upstream of GFP to develop a selection mini-gene (CrystallGFP) based on lens expression. The CrystallGFP mini gene is only 1.7 kb long and can be added to transgene constructs. To test the specificity of the promoter, we co-electroporated the CrystallGFP construct together with a ubiquitously expressed RFP (CAG-RFP) into the optic cup of a quail E3 embryo (*Figure 2A–D*). Twenty-four hours after electroporation, RFP-positive cells were found in the retina and lens (*Figure 2B,D*). However, only the lens cells expressed the GFP (*Figure 2C,D*), showing the specificity of the βB1crystallin promoter for lens tissues. Transgenic quails carrying the CrystallGFP selection cassette display strong expression of GFP in all lens cells during embryogenesis (*Figure 2E–G*) and in adults (*Figure 2I*). The CrystallGFP selection cassette was included in some of the transgenesis constructs, such as the muscle-specific quail line described below (*Figure 2H* and Figure 4).

## Generation of transgenic quails lines

### TgT2(CAG:NLS-mCherry-IRES-GFP-CAAX): ubiquitous expression of membranal GFP and nuclear RFP

To generate a transgenic quail line ubiquitously expressing a membrane-bound GFP and a nuclear RFP, embryos injected with the pT2-CAG:NLS-mCherry-IRES-GFP-CAAX plasmid (see above) were incubated until hatching and raised to adult stage. In this and other experiments described below, we have observed that about half of the (50) injected eggs hatched. Six weeks later, we collected semen from adult males and tested by PCR for the presence of the transgene. Three (F0) males, positive for the transgene, were crossed with four females each. From these crosses, three transgenic (F1) birds could be readily spotted at hatching by fluorescence screening thanks to the ubiquitously expressed GFP and mCherry. Expression of GFP or mCherry was visible in the beak, eyes and legs

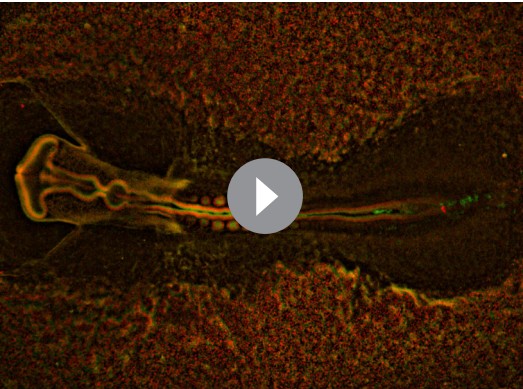

**Video 1.** Time-lapse video of an E2 TgT2(CAG:GFP-CAAX-IRES-NLS-mCherry) embryo observed in ovo. Embryo was maintained at 38˚C and imaged every 10mn for 12 hr using Thunder Imager Model Organism Leica stereo microscope equipped with 1x lens.
https://elifesciences.org/articles/56312#video1

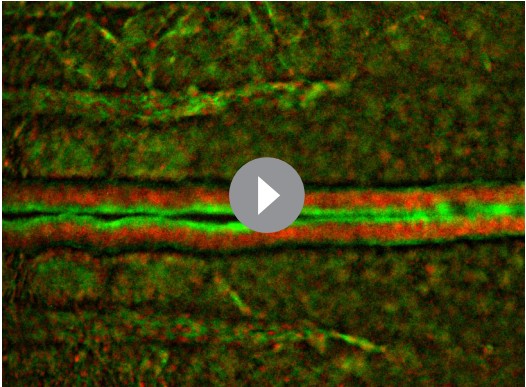

**Video 2.** Time-lapse video of an E2.5 TgT2(CAG:GFP-CAAX-IRES-NLS-mCherry) embryo. Embryo was maintained at 38˚C and imaged every 10mn for 12 hr using Thunder Imager Model Organism Leica stereo microscope equipped with 5x lens.
https://elifesciences.org/articles/56312#video2

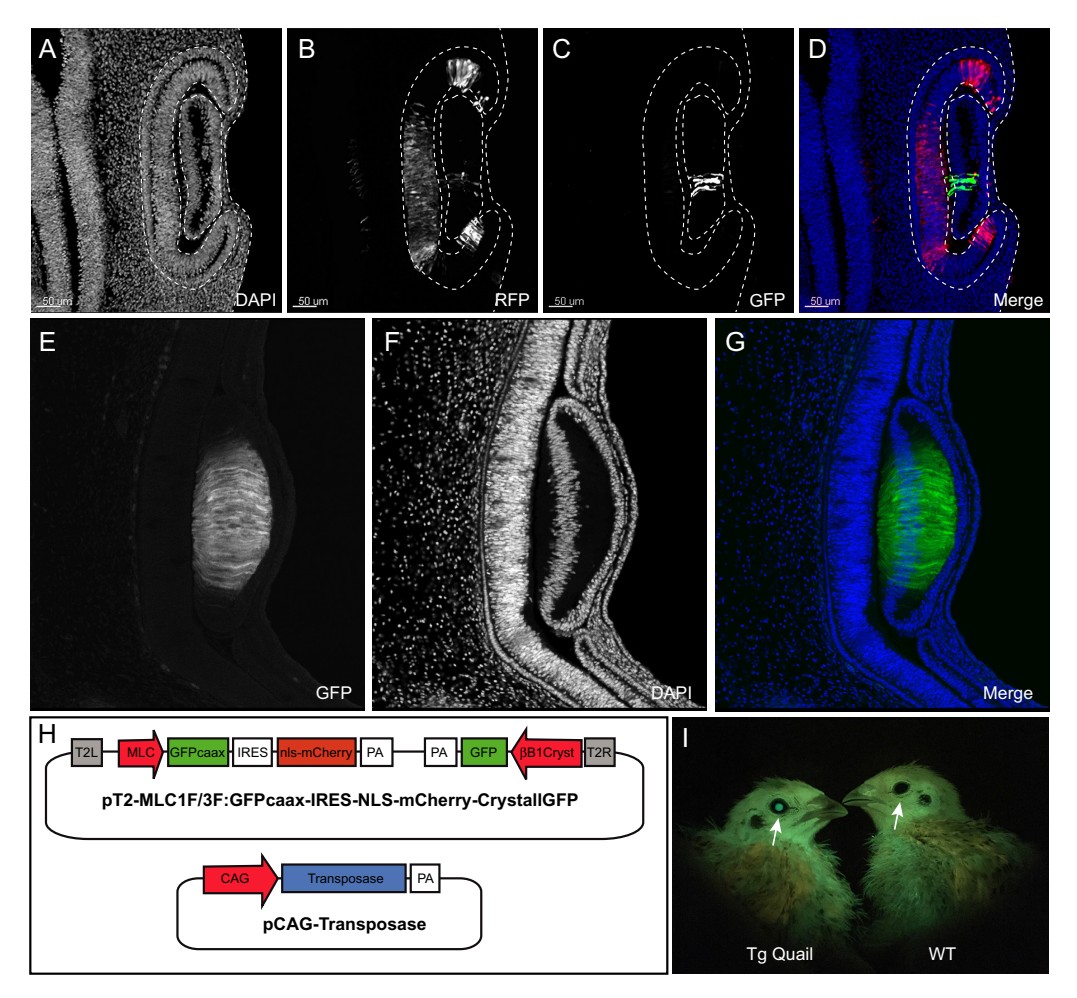

**Figure 2.** Design and use of the CrystallGFP mini gene. (**A–D**) Cross-section of the head of an E4 embryo, electroporated one day earlier in the optic cup with a CrystallGFP minigene. (**A**) DAPI, (**B**) electroporation marker CAG-RFP plasmid, (**C**) GFP, (**D**) overlay. (**E–G**) Cross-section of the head of a 3-day-old embryo of the *Tg(MLC:GFP-IRES-NLS-mCherry,CRYBB1:GFP)* transgenic line showing the specific expression of GFP throughout the lens. (**H**) Electroporation constructs used to express the CrystallGFP minigene in a muscle-specific transgenic line (see *Figure 4*). (**I**) Transgenic and WT adults of the muscle-specific transgenic line showing GFP expression in lens.

of the transgenic birds compared to wild-type animals (*Figure 1G,H*). Immunostaining on cross-sections of E3 transgenic embryos showed a ubiquitous expression of the GFP at the cell membrane and of mCherry in nuclei (*Figure 1I–K*). From this and other crosses we have performed in the laboratory (see below), we estimate that about 1% of the offspring contain the transgene, an efficiency comparable to that observed in the chicken using the same technology (*Tyack et al., 2013*). Compared to the existing quail lines carrying ubiquitously expressed fluorescent proteins, this line should prove useful to researchers in the field. Indeed, we observed that the membrane-bound GFP results in a better resolution of cell membrane processes (protrusions, filopodia, etc.) than a cytoplasmic counterpart, while it also combines a nuclear mCherry, allowing accurate segmentation of cells necessary for automated image analyses such as for 3D cell tracking. As a proof of concept of the usefulness of this transgene, we performed real-time video microscopy on 2-day-old embryos (observation time of about 12 hr), which illustrates the extensive morphogenetic changes taking place during early development (e.g. somitogenesis, heart and otic placode formation, etc.; see *Video 1*), while a higher magnification exquisitely shows the posteriorward migration of the pronephric primordium (see *Video 2*) in this embryo.

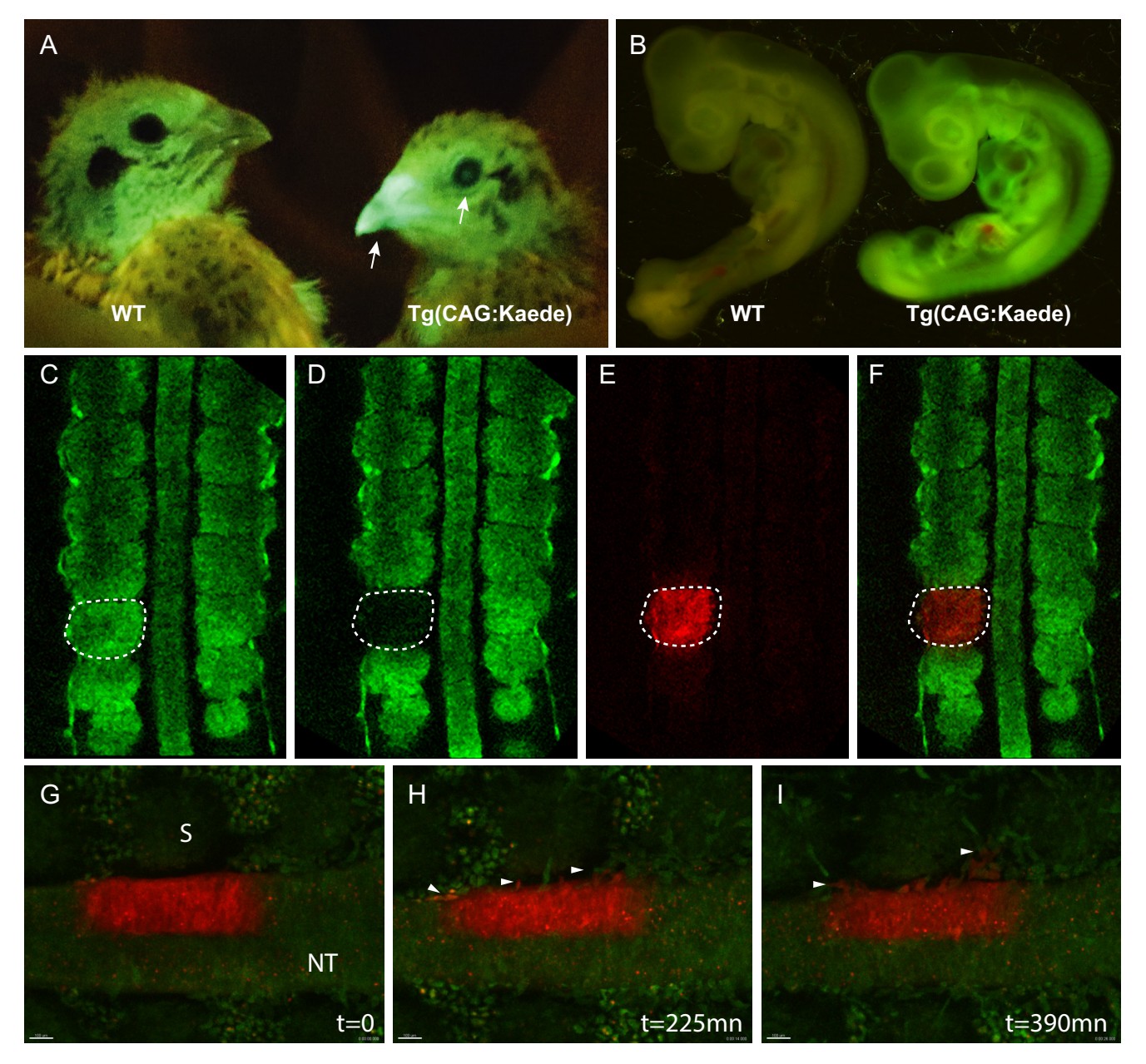

**Figure 3.** Generation of the photoconvertible Kaede transgenic quail line TgT2(CAG:Kaede). (**A**) Two-week-old WT and transgenic quails showing the ubiquitous expression of the green fluorescent Kaede in the beak and eye (arrows). (**B**) WT and transgenic 3-day-old embryos showing strong ubiquitous expression of the protein. (**C–F**) A newly formed somite before (**C**) and after (**D–F**) photoconversion. (**G–I**) Snapshots from a time-lapse video (see *Video 3*) showing the morphogenic movements of photoconverted neural tube cells. Arrowheads in H and I show neural crest cells initiating their lateral migration. NT: Neural Tube, S: Somite.

## TgT2(CAG:Kaede): ubiquitous expression of a photoconvertible fluorescent protein

To generate this transgenic line, E2 quail embryos were injected with a construct coding for a cytoplasmic form of the photoconvertible fluorescent protein Kaede (*Ando et al., 2002*), driven by a CAG promoter. Upon irradiation with ultraviolet light, Kaede undergoes irreversible photoconversion from green to red fluorescence. Three F1 founders were obtained in which strong and ubiquitous expression of the photoconvertible fluorescent protein is observed in adult (*Figure 3A*) and in

developing embryos (*Figure 3B*). Using the region of interest (ROI) function present in most confocal microscopes, specific areas of the embryo can be UV-illuminated to efficiently photoconvert the green fluorescent Kaede protein present in tissues to its red counterpart (*Figure 3C–F*). The long half-life of the photoconverted Kaede results in red fluorescence that can be detected up to 48 hr after photoconversion (*Tomura et al., 2008*). One major application of the TgT2(CAG:Kaede) quail line is the possibility to track in vivo the behaviour of cells over time. As an example, we performed a 7 hr-long time-lapse video of an E2 TgT2(CAG:Kaede) quail embryo where a section of the neural tube had been photoconverted upon exposure to UV light. Over the 7 hr of the time-lapse (one image taken every 15mn), neural crest cells can be observed migrating away from the neural tube (*Figure 2G–I*, arrowheads, and *Video 3*). This quail line is the first avian line carrying a photoconvertible fluorescent protein and it should be extremely useful to perform short to medium-term lineage tracing of cells as development proceeds.

## TgT2(*Mmu.MLC1F/3F:GFP-CAAX-IRES-NLS-mCherry,Gga.CRYBB1:GFP*): a skeletal muscle-specific reporter quail

We generated a line carrying a promoter for the mouse alkali Myosin Light Chain gene (MLC; *Kelly et al., 1995*) upstream of the membrane-bound GFP and the nuclear mCherry reporters described above. We designed a muscle-specific promoter, based on a synthetic reporter derived from the MLC1F/3F gene regulatory sequences previously utilized for mouse transgenesis (3F-nlacZ-E; *Kelly et al., 1995*). It contains a 2 kb sequence located 5' and 3' of the MLC3F transcriptional start site together with a 260 bp enhancer sequences from the 3' UTR region of the MLC3F gene, necessary for the high level of transcription in muscles. This construct was shown to drive strong LacZ expression in all (head and body) striated muscles from the early steps of myogenesis in somites of mouse embryos throughout embryogenesis, as well as in all skeletal muscles of the foetus and in the adult (*Kelly et al., 1995*). We included in the transgenesis construct the CrystallGFP cassette to facilitate the selection of F1 transgenic birds (*Figure 2H*).

F0 founder males were crossed with females and from 242 chicks that hatched, 3 transgenic F1 founders (1 male and 2 females, that is 1.2% efficiency) were selected, based on the expression of GFP in the lens (*Figure 2I*). These F1 were used to characterize the expression of the transgene during embryogenesis.

The GFP and RFP reporters were expressed in all (i.e. head, trunk and limb) skeletal muscles of the developing embryo (*Figure 4I–N*). On sections of E3 embryos, stained for GFP and RFP, strong expression was detected throughout the myotome of trunk somites (*Figure 4A–D*). We observed that the first sign of mCherry expression was detected within the transition zone (TZ; *Figure 4E–H*), where MYF5-expressing cells emanating from the medial border from the overlying dermomyotome translocate and extend along the antero-posterior axis of the embryo to form myocytes (*Gros et al., 2009*; *Gros et al., 2004*; *Rios et al., 2011*). This is coherent with a recent characterization of the MLC promoter we have done using in vivo electroporation in the chicken and where we found that its activity is initiated in myogenin-expressing, terminally differentiating myogenic cells within the TZ (*Sieiro et al., 2019*). MLC thus drives expression of the reporter genes from the early stages of myogenesis. It remains active in terminally differentiated myofibres, similar to what had been observed in 3F-nlacZ-E mouse embryos (*Kelly et al., 1995*). In contrast to this mouse line, where the expression of LacZ was observed in non-skeletal tissues (brain, optic vesicle and heart; *Kelly et al., 1995*), we did not detect the transgene in those tissues in the transgenic quails we analyzed, suggesting a more rigorous restriction to the

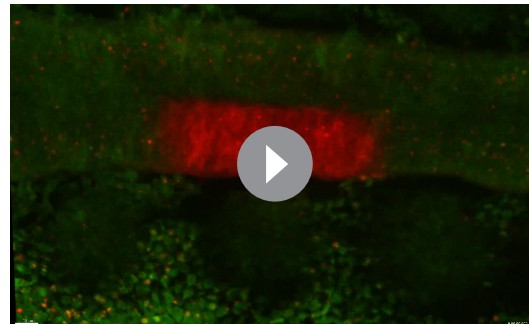

**Video 3.** Time-lapse video of an E2.5 TgT2(CAG: Kaede) embryo. Embryo was imaged using a Leica SP8 upright confocal microscope. A ROI was defined in half of the neural tube and exposed to UV light, photoconverting the Kaede protein from green to red. The area was imaged every 15mn for 7 hr showing neural crest cells migrating away from the neural tube.
https://elifesciences.org/articles/56312#video3

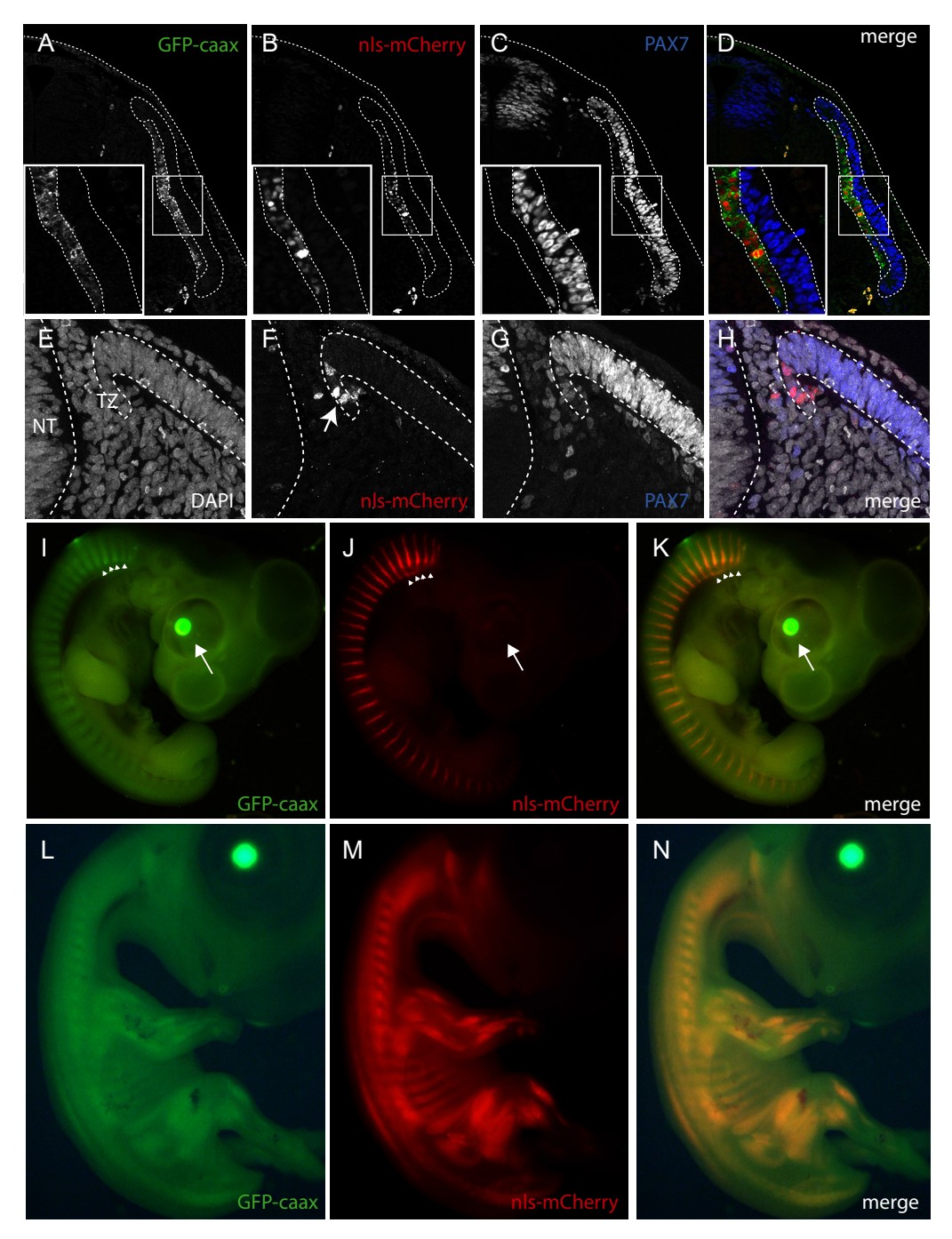

**Figure 4.** Description of muscle specific transgenic quail TgT2(Mmu.MLC1F/3F:GFP-CAAX-IRES-NLS-mCherry,Gga.CRYBB1:GFP). (A–D) Cross-section of E3 transgenic embryo stained for the indicated markers, showing the expression of the transgene throughout the primary myotome. (A) GFP-CAAX, (B) NLS-mCherry, (C) Pax7, (D) Merge. Insets in (A–D) Magnifications of the regions indicated in (A–D) showing the cellular localisation of the markers. (E–H) E5 Transgenic embryo showing GFP-CAAX (E) and NLS-mCherry (F) in the transition zone (TZ, arrow) where progenitors from the dermomyotome translocate to elongate and differentiate. (I–K) E5 embryos showing strong and specific expression of the muscle-specific reporter in somites (arrowheads). In this quail line, transgenic embryos can be selected at hatching by τηε GFP expression in lens due to the CrystallGFP minigene (arrows). (H–J) E7 transgenic embryo showing muscle-specific expression of the transgene in the head, limbs and trunk.

skeletal muscle lineage, an observation previously made in the chicken (*McGrew et al., 2010*). Refined details of the developing head, body and limb musculatures can be observed when (3DISCO) cleared E3 and E6 embryos are stained for GFP and RFP (together with an immunostaining against neural crest derivatives in blue for the E4 embryo) and imaged with light sheet microscopy (*Videos 4* and *5*). This quail line dedicated to skeletal muscles should be useful to all researchers in the field interested in characterizing the dynamics of myogenic differentiation during embryogenesis.

## Quailnet: a community website to share quail lines and resources
### Transgenic lines
To foster collaborative work in the avian community, we created a website in which existing and future quail lines will be listed (http://quailnet.geneticsandbioinformatics.eu/). A restricted access to the website enables researchers that generate new quail lines to deposit information (e.g. the type of line and the method used to generate it, its availability, etc.). An online form allows contacting the researcher that produced the line and inquire about additional information. Currently, a total of 24 lines are listed, comprising three transgenic lines generated using the transposon-based method (this report), seven transgenic lines generated using the lentiviral-based method (*Huss et al., 2015*; *Moreau et al., 2019*; *Saadaoui et al., 2020*; *Sato et al., 2010*; *Seidl et al., 2013*), 7 quail mutant lines and 7 strains obtained through breeding-selection programs (See *Figure 5A–C*).

## Gene and genomic information
The Quail Genome Consortium has recently obtained high quality genomic data, assembled and submitted for annotation at NCBI and Ensembl (*Morris et al., 2019*). This is a critically important step for the future design of genome engineering technologies in this organism, such as the Crispr-Cas9-based gene knockout and knock-in. QuailNet has been fitted with a gene search feature, which provides useful information such as gene models, curated coding sequences (not available elsewhere) and a genomic browser (*Figure 5D,E*). Furthermore, each result page embeds a link to the protocol for designing efficient Crispr-Cas9-mediated gene knock-out based on our own recent experience (*Morin et al., 2017*; *Véron et al., 2015*) and a link to the ChopChop (*Labun et al., 2019*) website that we found helpful and user-friendly for the choice of gRNAs sequences.

## Additional resources
As an aid to transgenic quail research, QuailNet integrates various resources:

- The 3D quail anatomy portal, which contains 3D models of quail embryos covering a range of development stages from embryonic day E1 (HH7) to E11 (HH40).

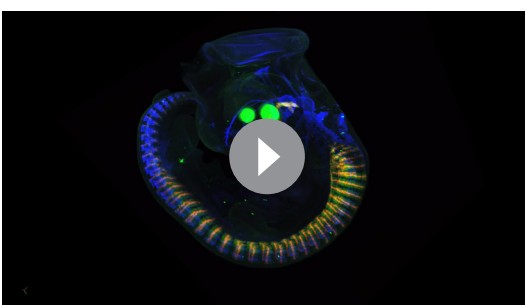

**Video 4.** Quail TgT2(Mmu.MLC1F/3F:GFP-CAAX-IRES-NLS-mCherry,Gga.CRYBB1:GFP) embryo at 3 days of development, immunostained for GFP (green) and mCherry (red) from the transgene and counterstained for neural crest (HNK1, blue), clarified by the '3DISCO' technique and imaged with a light sheet microscope (LSFM Z1 Zeiss). Image rendering and video obtained with an Arivis software suite.
https://elifesciences.org/articles/56312#video4

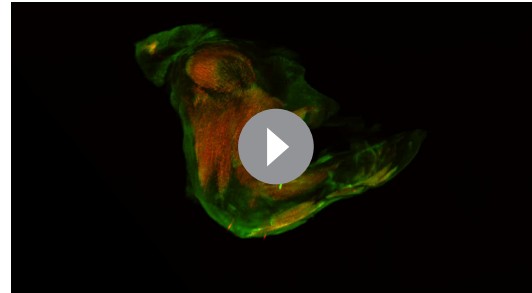

**Video 5.** Wing of a quail TgT2(Mmu.MLC1F/3F:GFP-CAAX-IRES-NLS-mCherry,Gga.CRYBB1:GFP) embryo at 6 days of development, immunostained for GFP (green) and mCherry (red), clarified by the '3DISCO' technique and imaged with a light sheet microscope (LaVision Biotec). Image rendering and video obtained with an Arivis software suite.
https://elifesciences.org/articles/56312#video5

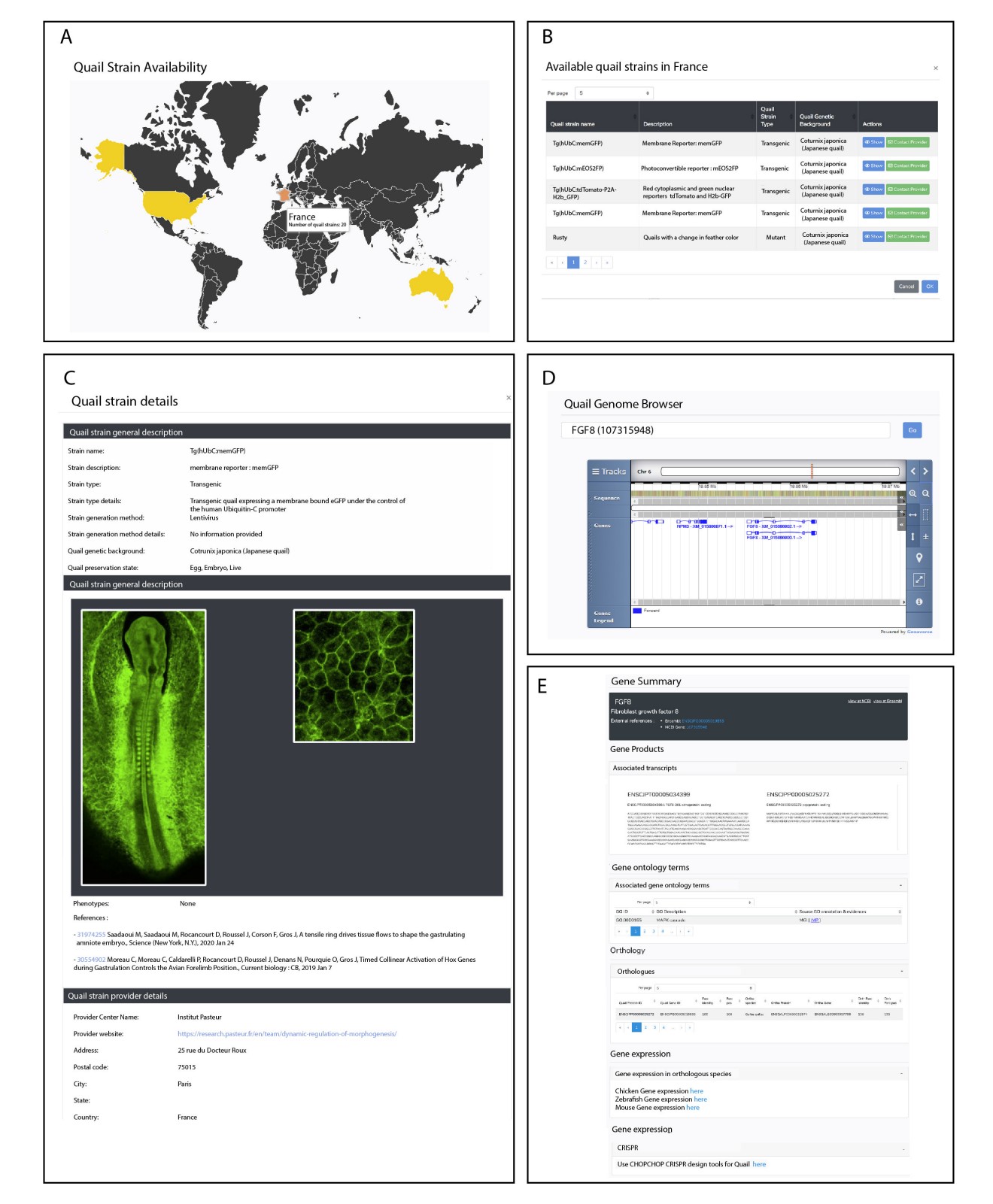

**Figure 5.** Description of QuailNet features. (**A**) Interactive world map displaying the number of quail strains available by country. (**B**) List of quail strains together with a general description by country. (**C**) Detailed description of a specific quail strain (e.g. Tg(hUbC:memGFP)). (**D**) Quail genome browser displaying genomic information and location of a queried gene (e.g. FGF8). (**E**) Information associated with a queried gene (e.g. FGF8).

- When available, links to the chicken embryo gene expression database GEISHA (*Antin et al., 2014*) is provided on the gene result page.
- Useful protocols that relate to data generated with transgenic animals.

Altogether these genomic, molecular genetics, and bioinformatics resources aim at elevating the quail to the rank of genetic laboratory animal model of reference for avians.

## Discussion

The avian animal model has been historically important in various areas of biomedical research and in particular in the developmental biology field. The easy access to the embryo has fostered the development of a highly diverse range of techniques to track cells, manipulate their environment and modulate their gene function that, combined with in vivo time-lapse imaging, has brought crucial insights into many dynamic biological processes. However, the somewhat technically demanding aspects of those techniques have hampered their widespread use by many laboratories. The possibility to easily generate genetically modified bird lines would represent a game changer in the field that may incite established researchers and newcomers alike to consider using genome engineering in birds to address their questions. While the generation of transgenic quails using lentiviruses has met very significant successes, we believe that the technique of liposome-mediated PGC transfection that we describe in this report, with its simplicity, efficacy and extended possibilities of diversifications, will serve as a landmark in the use of quail as an economic alternative to others vertebrate models. In the lines generated here, the size of the transgenic DNA constructs inserted between the two Tol2 sites ranged from 3 kb for the Kaede photoconvertible line to 7.5 kb for the MLC muscle specific line. This flexibility thus allows the design of complex constructs and the use of large tissue-specific promoters that would be nearly impossible to accommodate in lentiviruses.

The website we have generated completes this technical advance and provides a wealth of information and resources that should significantly help future developments of the technology, including for the targeted insertion (knock-in) of foreign DNA into the quail genome.

A significant step to promote genetic approaches in avian is the recent creation of a quail transgenic facility (MQTF) at Monash University, where the quail lines described in this report were established and novel technologies are being developed at present. At MQTF (see link to website on QuailNet), transgenic birds can be made-to-order for avian researchers around the world. The worldwide availability of existing and future quail lines is currently addressed and details about this can be found in the QuailNet website. It is our belief that the creation of this facility constitutes the cornerstone of a network of future avian facilities that will strengthen our community and attract new generations of researchers across fields to one of the most versatile experimental systems there is.

## Materials and methods

### Generating transgenic quail by direct injection

The direct injection technique is performed as described in *Tyack et al., 2013*. Plasmid DNA was purified using Nucleobond Xtra Midi EF kit. Injection mix contained 0.6 µg of Tol2 plasmid, 1.2 µg of CAG Transposase plasmid, 3 µl of Lipofectamine 2000 CD (ThermoFisher Scientific) in 90 µl of Opti-Pro transfection medium. 1 µl of injection mix was injected in the dorsal aorta of 2.5-day-old embryos. After injection, eggs were sealed and incubated until hatching. Chicks were grown for 6 weeks until they reached sexual maturity. Semen from the male was collected using a female teaser and massage technic as described in *Chełmońska et al., 2008*. In short, the foam produced by the male cloacal gland was first emptied by pressing the gland. The male was then introduced in a cage, which already contained a female. When the male was ready to mate, it was taken out, turned on its back and a massage of the cloaca led to the expulsion of semen that was collected. Genomic DNA from semen was extracted and PCR was performed to test for the presence of the transgene. Males showing a positive band were kept and crossed with wild type females. F1 offsprings were selected directly after hatching using UV goggles if expression of the transgene was readily visible in newly hatched chicks or by genotyping 5 days after hatching by plucking a feather.

## Expression constructs

The T2(CAG:NLS-mCherry-IRES-GFP-CAAX) has been described previously (*Sieiro-Mosti et al., 2014*). The T2(CAG:Kaede) was constructed by cloning the Kaede fluorescent protein (*Ando et al., 2002*) into a the T2(CAG) expression vector. The T2(Mmu.MLC1F/3F:GFP-CAAX-IRES-NLS-mCherry, Gga.CRYBB1:GFP) was made by combining two constructs. The first is the mouse Myosin Light chain MLC1F/3F:GFP-CAAX-IRES-NLS-mCherry, described in *Sieiro-Mosti et al., 2014*. The second was made by PCR amplification from chicken genomic DNA of a 462bp-long promoter region of the chicken CRYBB1 gene (*Duncan et al., 1995*). As indicated in *Figure 2H*, both constructs are cloned in a head-to-tail configuration as we observed that this minimizes interferences between promoters.

## Section, immunochemistry and confocal analysis

Transgenic embryos were dissected and fixed for 1 hr in 4% formaldehyde. For cryostat section, embryos were embedded in 15% sucrose/7.5% gelatine/PBS solution and sectioned at 20 µm slices. The following antibodies were used: anti-GFP chicken polyclonal (Abcam), anti-RFP rabbit polyclonal (Abcam), anti-Pax7 IgG1 mouse monoclonal (Developmental Studies Hybridoma Bank), anti-Vasa (gift from Dr Craig Smith laboratory). Stained sections were examined using a Leica SP5 confocal microscope 40x lens oil immersion and images were analyzed with an Imaris software suite.

## Quail lens electroporation

Lens electroporation were performed as described in *Chen et al., 2004*. Plasmids were electroporated at 1 µg/µl final concentration in the electroporation mix. Lens electroporation were performed in 2-day-old embryos (HH12) by positioning the electrodes to target the lens. Embryos were re-incubated at 38°C for 24 hr.

## Embryo clearing, staining and imaging

Embryo preparation for whole mount immunostaining and 3DISCO clearing was performed as described in *Belle et al., 2017*. Imaging of stained embryos was performed (for the E3 quail embryo) on a Zeiss Lightsheet Z1 microscope equipped with 5X Plan-Neofluar objectives and (for the E6 quail wing) on LaVision Biotec Ultramicroscope II. Image rendering was performed with an Arivis software suite. The following antibodies were used: anti-GFP chicken polyclonal (Abcam), anti-RFP mouse IgG1 monoclonal (Abcam), anti HNK1 mouse IgM monoclonal (Developmental Studies Hybridoma Bank).

## Time lapse imaging

Live imaging was performed in ovo using a custom-made egg incubator designed for live observation as described in quailDB (http://quailnet.geneticsandbioinformatics.eu/). The transgenic quail eggs were carefully placed in a stainless-steel cup without damaging the egg yolk. Embryo turn to the top and the stainless-steel cup is filled up with egg white. A CultFoil 25 µm Teflon membrane (Zeiss, allowing gas exchange and avoiding dehydration) was placed over the embryo. The stainless-steel cup is then placed on a heat pad, which maintains the temperature of the embryo at 38°C. Embryos were imaged using a Leica Thunder Imager Model Organism (*Videos 1* and *2*) or under a Leica SP8 confocal upright (*Videos 3* and *4*).

## Nomenclature

We propose to describe the quail transgenic lines we generated according to the nomenclature conventions described in the ZFIN zebrafish website (https://wiki.zfin.org/display/general/ZFIN+Zebra-afish+Nomenclature+Conventions). For instance, in the line *TgT2(Mmu.MLC1F/3F:GFP-CAAX-IRES-NLS-mCherry,Gga.CRYBB1:GFP)*, *Tg* denotes transgenic, *T2* denotes the Tol2 transposons, *Mmu* and *Gga* denotes the species of origin of the two promoters used in the transgene (Mmu = *Mus musculus* Gga = Gallus gallus).

## Acknowledgements

The authors thank Terry wise, Chris Darcy and Mark Tizard from CSIRO AAHL (Geelong Australia) for their expertise in direct injection and transgenic chicken breeding. The authors acknowledge

Monash Micro Imaging, Monash University, for the provision of instrumentation, training and technical support. The Australian Regenerative Medicine Institute is supported by grants from the State Government of Victoria and the Australian Government. We thank the Faculty of Medicine and Health Science and Monash Technology and Research Platforms for their financial support. MQTF was supported by Faculty Strategic Grants Schemes from Monash University to OS and CM. CM and MJD were supported by grants from Stem Cells Australia (CSA) and the Association Française contre les Myopathies (AFM).

## Additional information

### Funding

| Funder | Grant reference number | Author |
| --- | --- | --- |
| Association Française contre les Myopathies | Research grant | Christophe Marcelle |
| Stem Cells Australia | Research grant | Christophe Marcelle |

The funders had no role in study design, data collection and interpretation, or the decision to submit the work for publication

### Author contributions

Olivier Serralbo, Conceptualization, Data curation, Formal analysis, Validation, Investigation, Visualization, Writing - original draft, Project administration; David Salgado, Conceptualization, Resources, Software; Nadège Véron, Caitlin Cooper, Conceptualization, Investigation; Marie-Julie Dejardin, Investigation, Visualization; Timothy Doran, Conceptualization, Methodology; Jérome Gros, Conceptualization, Resources, Software, Methodology, Writing - original draft; Christophe Marcelle, Conceptualization, Supervision, Funding acquisition, Writing - original draft, Project administration, Writing - review and editing

### Author ORCIDs

Olivier Serralbo (iD) http://orcid.org/0000-0003-0808-3464
Christophe Marcelle (iD) https://orcid.org/0000-0002-9612-7609

### Ethics

Animal experimentation: All procedures were approved by a Monash University Animal Ethics Committee (ERM ID 15002, ERM ID 18809) in accordance with the Australian Code for the Care and Use of Animals for Scientific Purposes (8th Edition, 2013).

### Decision letter and Author response

Decision letter https://doi.org/10.7554/eLife.56312.sa1
Author response https://doi.org/10.7554/eLife.56312.sa2

## Additional files

### Supplementary files

• Transparent reporting form

### Data availability

All data generated or analysed during this study are included in the manuscript and supporting files.

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
