## [Decision Letter]

**Acceptance summary:**

The manuscript describes the generation of three new transgenic quail lines using a relatively simple lipofection method using transposon-based plasmids (previously described) in quail embryos in ovo, by injecting the corresponding DNAs in the dorsal aorta and targeting circulating primordial germ cells. The three transgenic lines include: an ubiquitous double fluorescent line, a Kaede-based photoconvertible line and a skeletal muscle-specific line. The manuscript also describes the generation of a webpage (the Quail net Project) where different quail transgenic lines generated by different groups are listed together with direct access to curated quail sequences and other resources. In summary, it provides the avian community with new potent tools and encourage the use of quail as a solid genetic model.

**Decision letter after peer review:**

Thank you for submitting your article "Transgenesis and web resources in quail" for consideration by *eLife*. Your article has been reviewed by three peer reviewers, and the evaluation has been overseen by Marianne Bronner as the Senior and Reviewing Editor. The following individuals involved in review of your submission have agreed to reveal their identity: Claudio D Stern (Reviewer #1); Cornelis Weijer (Reviewer #2); Aixa Victoria Morales (Reviewer #3).

The reviewers have discussed the reviews with one another and they all were enthusiastic about the resource. However, they point out areas in which the manuscript would require further information and clarification, all of which I believe can be achieved by textual changes. I refer you to the complete reviews below for details. Given the current world situation, we will give authors as much time as they need to submit revised manuscripts.

Reviewer #1:

This is a very nice paper submitted as a "Tools and Resources" type of paper. It reports a very simple method for generating transgenic quails. The method consists of just injecting a mixture of the plasmid together with lipofectamine into the circulating blood stream of quail embryos at about E2.5, where the construct transfects some of the circulating primordial germ cells (PGCs). An elegant feature is a transgene that drives GFP expression in the lens (based on an enhancer of the crystallin gene) which allows quick and easy identification of transgenic animals in F2 for crossing and generating stable lines. The efficiency of transgenesis appears to be very high and the method lends itself to knock-in and knock-out methods for example by using CRISPR-Cas9 technology in combination with this transfection method. In principle this technique should be easily achievable in any lab with access to animal house facilities that can maintain quails (which need relatively little space and reach sexual maturity quickly). In the paper, several transgenic lines are generated including a nuclear RFP + membrane GFP double transgenic, a photoconvertible Kaede line for cell tracing, and a reporter for muscle. Examples of time-lapse films and 3d confocal imaging illustrate the usefulness of some of these for high resolution imaging and a variety of experiments. Finally the paper reports a set of online resources integrated with other existing databases, including a repository of available transgenic lines, a genome browser and other features.

Overall the paper is very well written and very nicely illustrated. I have no substantial comments to make and I feel that the paper can be published essentially "as is", without the need for revision.

Reviewer #2:

In this paper it is shown that the in vivo transformation of primordial germ cells developed in chick can be adapted to quail. This makes it due to its short 6 week lifecycle and small size highly suitable for avian transgenesis experiments. The transgenesis efficiency (~1%) is manageable, especially when combined with rapid screening techniques, including such as the here developed eye lens specific crystallGFP. The authors describe the generation of three novel transgenic quail lines, a membraneGFP and nuclear RFP marker (TgT2(CAG:NLS-mCherry-IRES-GFP-CAAX)), a muscle specific marker using the promoter of the alkali myosin light chain (TgT2(Mmu.MLC1F/3F:GFP-CAAX-IRES-NLS-mCherry,Gga.CRYBB1:GFP)) as well as photo-convertable GFP line (TgT2(CAG:Kaede)).

The paper concludes with a section outlining a web resource bundling some key information concerning quail related research (transgenic lines, genomics, expression and anatomy databases as well as some tools), which useful, however see comments below

The paper is mainly a method/resource paper, it is clearly written and the methods/observations are well supported and documented. The videos are an effective addition to the figures and text. The information will be of great value to the community.

Points that would be helpful to address in more detail are:

Assuming that it may prove difficult for most researchers to set up their own quail colonies and long distance shipping may not be a realistic option, effective use of these transgenic lines will require availability through a local providers. It would be useful to outline whether this is/can be addressed.

The website at the moment primarily summarises the currently existing transgenic quail lines and some novel information on anatomy. It furthermore bundles other web resources (genomics, expression studies etc., some links not active).

In order to extend this to a real community resource it will be critical to know how it is going to be maintained and updated in the longer term. Can data be provided to the anatomy section/Materials and methods section? How is this going to be curated? etc.

Reviewer #3:

In general, the manuscript is well written and clearly presented with beautiful and meaningful images and videos. However, there are some aspects that should be improved:

1) The manuscript does not give any information concerning the number of the transgene copies that were inserted in each of the generated lines or how many separated F1 founders they were checking for each of the three transgenic lines described. It is not clear either, how stable will be the transgenic lines after several generations, as they only give data on the F2.

2) The manuscript discuss poorly the transposon method they have used in relation to other methods such as lentivirus or adenovirus based method in relation to the efficiency of germline transmission. For instance, adenovirus mediated CRISPR/Cas9 mediated deletion has a transmission between 2.5 to 10% in a recent article by Lee et al., 2019, or in another recently lentivirus generated line, Ubiquitous expression of membrane tethered EGFP generated by one of the labs in the manuscript, rendered a transmission of 8% (Saadaoui et al., 2020).

3) The authors should provide more details about the protocol for transgenesis: DNA quality purification, quail sperm obtaining, specific quail embryos clearing for 3DISCO.

---

## [Author Response]

Reviewer #2:[…] Points that would be helpful to address in more detail are.Assuming that it may prove difficult for most researchers to set up their own quail colonies and long distance shipping may not be a realistic option, effective use of these transgenic lines will require availability through a local providers. It would be useful to outline whether this is/can be addressed.

The worldwide availability of the lines we generated and the ones to come is obviously an important issue that needs to be addressed if we want the technique to spread in the scientific community. Since transgenesis in quail is a rather recent technology, we can only talk about how a handful of colleagues that contacted us addressed this question so far. Few decided to set up small quail husbandries to maintain quail lines locally. Others decided to set up collaborations with the laboratories that generated and maintain the lines, which regularly provide fertilized eggs for their experiments. While setting up a quail husbandry locally may sound attractive, the amount of paperwork required by local and national administration, ethics and GMO committees, can be a significant deterrent as there are no established regulations on transgenesis with this novel animal model. Collaborations, on the other hand, are usually done on a small scale, and within reasonable distances from the source. To add to the confusion, international importation laws also forbid the transport of fertilized eggs between certain countries. To address those concerns, we are working on establishing reliable sources of fertilized eggs in two locations. The first is at the Veterinary school of UC Davis, CA, with whom a verbal agreement has been reached in recent months to centralize some of the existing quail lines and distribute them in the US; the second is in France, where ongoing discussions with the Lyon Veterinary School aim for something similar in Europe. Those consultations involve quite a bit of persistence and their outcomes are still uncertain. In this evolving situation, we cannot give specific details in the manuscript that address this issue, but we propose to mention those details in the QuailNet Website and refer to them in the manuscript as follows: "The worldwide availability of existing and future quail lines is currently addressed and details about this can be found in the QuailNet website".

The website at the moment primarily summarises the currently existing transgenic quail lines and some novel information on anatomy. It furthermore bundles other web resources (genomics, expression studies etc., some links not active).In order to extend this to a real community resource it will be critical to know how it is going to be maintained and updated in the longer term.

The website has been designed in two parts: (i) a publicly accessible part (the website) and (ii) a section restricted to administrators and resource providers. Administrators will be in charge of populating content and curating submitted data. Regarding genomics, orthologues, expression data, etc., automatic scripts have been designed to ensure the regular updating of databases with the latest available information. The website is currently under a development phase and it is hoped that it will evolve as the technology spreads within the community. Major developments or changes to the existing website would however require specific fundings for a developer's salary. Upon request, the scripts/database structure can be released through a git repository so that anyone can contribute.

Can data be provided to the anatomy section/Materials and methods section? How is this going to be curated? etc.

Regarding anatomy sections, this is part of a project called http://anatomyportal.org/, led by Dr. Robert Bryson Richardson and for which Dr. David Salgado was the lead developer. Please contact Dr. Richardson (robert.bryson-richardson@monash.edu) for updates or modifications to the website.

Reviewer #3:In general, the manuscript is well written and clearly presented with beautiful and meaningful images and videos. However, there are some aspects that should be improved:1) The manuscript does not give any information concerning the number of the transgene copies that were inserted in each of the generated lines or how many separated F1 founders they were checking for each of the three transgenic lines described. It is not clear either, how stable will be the transgenic lines after several generations, as they only give data on the F2.

We have not tested the actual number of copies in the genome of our founders, but the transmission of the transgene to the offspring in each of the quail lines indicates that the transgenes are present in one copy. This is coherent with the observation of our collaborator (T. Doran, that has developed this technology in chicken), which showed that among the 150 founders they analysed, a large majority (64%) contained one insertion in the chicken genome, while 27% contained two insertions.

These are the number of F1 founders for each transgenic line.

TgT2(CAG:NLS-mCherry-IRES-GFP-CAAX): 3 F1 foundersTgT2(CAG:Kaede): 3 F1 foundersTgT2(Mmu.MLC1F/3F:GFP-CAAX-IRES-NLS-mCherry,Gga.CRYBB1:GFP): 3 F1 founders

This has been added to the text.

We have not observed any significant decrease of the signal over time, suggesting, as mentioned in the text that Tol2 insertions are not particularly prone to inactivation over time.

2) The manuscript discuss poorly the transposon method they have used in relation to other methods such as lentivirus or adenovirus based method in relation to the efficiency of germline transmission. For instance, adenovirus mediated CRISPR/Cas9 mediated deletion has a transmission between 2.5 to 10% in a recent article by Lee et al., 2019, or in another recent lentivirus generated line, Ubiquitous expression of membrane tethered EGFP generated by one of the labs in the manuscript, rendered a transmission of 8% (Saadaoui et al., 2020).

The comparative efficiencies of those techniques is not really a limiting factor. To take an example, if a F0 male, found positive for the transgene with our technique, is crossed with 4 females that each lay an egg-a-day, one F1 will be made every 25 days of crossing (for an efficiency of 1%). As we usually cross 3 positive F0 males, we obtain about 3 F1 in the same period. The efficiencies that have been published about viruses are definitely higher (lentivirus: 8%; adenovirus: 2.5-10% efficiency) and they could allow a shorter crossing period, but in fact, our colleagues that use viruses tend to cross males with females for just about as long a period, such that they obtain more F1 than we do, which they anyhow cull to keep only two to three. Such that practically, there is not a huge advantage at having higher efficiencies to perform transgenesis. A low efficiency may become an important bottleneck if more delicate technologies are envisaged (e.g. KI), but it is unlikely that this can be solved with viruses, since the combined sizes of the Cas9, the gRNA, their respective promoters, the homology arms, and the KI construct (e.g. GFP) are clearly out of size range for viruses. As explained in the text, there are other advantages of the plasmid technique over the virus approach that make this technique attractive for specific applications.

3) The authors should provide more details about the protocol for transgenesis: DNA quality purification, quail sperm obtaining, specific quail embryos clearing for 3DISCO.

Plasmids are prepared using the Nucleobond endotoxin-free midiprep kit. This is now indicated in the Materials and methods section.

Semen is collected from 7 week-old males using a female teaser as described in

Chelmonska et al., 2008. More details about the technique have been added to the text.

As indicated in the Materials and methods section, we followed the exact protocol of the 3DISCO procedure described in the Cell paper of Alain Chedotal's team (Belle et al., 2017) to prepare the quail embryos for IHC and 3D imaging.